# A Comparative Study on Laser Powder Bed Fusion of Differently Atomized 316L Stainless Steel

**DOI:** 10.3390/ma15144938

**Published:** 2022-07-15

**Authors:** Krzysztof Grzelak, Marcin Bielecki, Janusz Kluczyński, Ireneusz Szachogłuchowicz, Lucjan Śnieżek, Janusz Torzewski, Jakub Łuszczek, Łukasz Słoboda, Marcin Wachowski, Zenon Komorek, Marcin Małek, Justyna Zygmuntowicz

**Affiliations:** 1Faculty of Mechanical Engineering, Institute of Robots & Machine Design, Military University of Technology, 2 Gen. S. Kaliskiego St., 00-908 Warsaw, Poland; krzysztof.grzelak@wat.edu.pl (K.G.); ireneusz.szachogluchowicz@wat.edu.pl (I.S.); lucjan.sniezek@wat.edu.pl (L.Ś.); janusz.torzewski@wat.edu.pl (J.T.); jakub.luszczek@wat.edu.pl (J.Ł.); marcin.wachowski@wat.edu.pl (M.W.); 23D LAB Sp. z o.o. Farbiarska 63B St., 02-862 Warsaw, Poland; marcin.bielecki@3d-lab.pl; 3Faculty of Materials Engineering, Institute of Materials Engineering, Military University of Technology, 2 Gen. S. Kaliskiego St., 00-908 Warsaw, Poland; zenon.komorek@wat.edu.pl; 4Faculty of Civil Engineering and Geodesy, Institute of Civil Engineering, Military University of Technology, 2 Gen. S. Kaliskiego St., 00-908 Warsaw, Poland; marcin.malek@wat.edu.pl; 5Faculty of Materials Science and Engineering, Warsaw University of Technology, 141 Woloska St., 02-507 Warsaw, Poland; justyna.zygmuntowicz@pw.edu.pl

**Keywords:** additive manufacturing, powder bed fusion, 316L stainless steel, ultrasonic atomization, gas atomization

## Abstract

The significant growth of Additive Manufacturing (AM), visible over the last ten years, has driven an increase in demand for small gradation metallic powders of a size lower than 100 µm. Until now, most affordable powders for AM have been produced using gas atomization. Recently, a new, alternative method of powder production based on ultrasonic atomization with melting by electric arc has appeared. This paper summarizes the preliminary research results of AM samples made of two AISI 316L steel powder batches, one of which was obtained during Ultrasonic Atomization (UA) and the other during Plasma Arc Gas Atomization (PAGA). The comparison starts from powder particle statistical distribution, chemical composition analysis, density, and flowability measurements. After powder analysis, test samples were produced using AM to observe the differences in microstructure, porosity, and hardness. Finally, the test campaign covered an analysis of mechanical properties, including tensile testing with Digital Image Correlation (DIC) and Charpy’s impact tests. A comparative study of parts made of ultrasonic and gas atomization powders confirms the likelihood that both methods can deliver material of similar properties.

## 1. Introduction

The most popular powders on the market for additive manufacturing come from gas atomization (PAGA), which can deliver large spherical powders [1,2]. This kind of technology facilitates obtaining a high amount of proper quality powders that could be successfully used in the AM of metallic powders in Powder Bed Fusion (PBF) and (DED) technologies. Even though the growth of AM has led to the initiation of many new research paths, the need to use low amounts of metallic powder for some highly specified solutions still exists [3,4,5]. In the case of gas atomization devices, a lot of practical issues arise when the material type needs to be frequently changed (i.e., steel after copper alloys). One alternative and novel powder production method is based on ultrasonic atomization (UA). Such a method is commercially available on the ATO system (3D Lab Ltd., Warsaw, Poland) and utilizes capillary waves at the ultrasonic frequency to break molten metal into fine droplets [6,7,8].

UA is one of the most promising technologies for powder production dedicated to AM and coatings at a lower cost than PAGA, especially in the production of smaller amounts of powders (batches 1–100 kg) of customized composition (special application steels, nickel/cobalt superalloys, titanium-aluminide inter-metallics, memory shape alloys, bulk metallic glasses, high entropy alloys, noble metals, and others). UA technology facilitates obtaining spherical particles with a targeted diameter from 15 to 150 μm. Due to the selection of optimal ultrasonic frequency, the UA method can deliver a powder batch in a customized diameter range, usually ordered either in diameters of 20–60 μm, 50–120 μm, or any other narrow ranges on request. Producing powder at high efficiency in targeted diameter ranges is a crucial advantage when compared to PAGA, as deliberated later. In general, PAGA methods are characterized by the production of material particles ranging from 15 to 300 μm; classification on sieves allows for a yield of ~30% of powder in diameters needed by the AM market.

While obtaining properly shaped powder in a specified diameter range is important, it is essential to assure the proper quality of the powder. The quality control plan should address, first of all, process stability [9,10] with the lowest possible porosity [11,12], which later enables the manufacturing of mechanical parts with satisfactory mechanical properties [13,14,15,16], including fatigue properties [17,18,19]. The obtainment of proper particle diameters during UA processing is driven by the ejection of fine droplets of liquid metal from an ultrasonic horn (sonotrode) through ultrasonic vibrations supplied to the molten metal pool. The thermal condition in this pool is kept above liquidus temperature by continuous melting with an electric arc. The UA process is executed in argon (99.999% purity), ventilating the atomization chamber with an inlet temperature of 20 °C. To better understand the process, the main operation principles are pointed out below:Raw material in the form of wire, rod, or pellets (optionally, from scrapped and milled AM print-outs “re-powder”) is continuously supplied to the pressurized atomization chamber via the pushing of the material toward the sonotrode hot end, where the atomization will take place.An electric arc is established between a non-consumable electrode and the sonotrode in order to melt the raw material on the sonotrode hot end to form a molten metal pool.The ultrasonic vibrations are transferred through the sonotrode from its cold end, where the ultrasonic transducer is assembled, toward the hot end, and eventually to the molten metal pool. Consequently, capillary waves are formed on its surface at the same frequency as determined by the transducer. Once the magnitude of the vibrations in the pools is enough to overcome the resistance forces of the surface tension and viscosity, the capillary waves start to become unstable, and some of their crests eject the droplets at a diameter dependent on the ultrasound frequency, surface tension, and liquid metal density.The droplets are ejected with some kinetic energy into the stream of the cooled inert gas. In proximity to the electric arc, the gas temperature is high enough that droplets are kept in a liquid state for a brief period, until the surface tensions round them off to almost perfect spheres.Further, the droplets cool down by convection and radiation processes against the inert gas, and then solidify. The particles are conveyed with the aid of aerodynamic forces toward the atomization chamber outlet.The stream of the warm inert gas with powder undergoes separation of the powder from the gas in a cyclone. Then the powder is collected below the cyclone in a sealed container.The gas from the cyclone is filtered of dust (particles of a few microns), cooled, and recirculated to the process in the previous step.The cooled powder is classified on various sieves to the particle sizes needed, e.g., in the range of 20–63 μm, as utilized in trials described in this paper.

In a continuous process, steps 1, 2, and 3 are run practically in the same place, i.e., the wire or rod is pushed to the sonotrode hot end. At the same time, nonstop melting keeps the molten metal pool on the sonotrode hot end, and then the ultrasonic vibrations eject the droplets from the pool—as is shown in Figure 1.

Due to the ordered nature of the capillary waves driven by the ultrasonic vibrations, the output power has a narrow particle size distribution, closely dependent on the chosen ultrasound frequency. For example, for any steel, the ATO system at a frequency of 35 kHz yields powder with D50 in the range of 45–50 μm. Some minor variation depends on other factors, e.g., the heating rate (a stronger electric arc allows a higher temperature of liquid metal, hence generating finer droplets at lower viscosity and surface tensions). Due to the application of a precisely calibrated ultrasonic system and optimized electric arc heating, which allows for the creation of a relatively uniform thermal flux toward the molten metal pool, most of the powder batch has diameters close to those desired by the AM market. This physics-based phenomenon targets powder size and produces a batch with a significantly higher utilization (conversion rate) than typical gas atomization.

The main aim of this paper is to compare the UA and PAGA powders’ properties, and produce sample parts with their use. Such a comparison was made to deeply analyze the differences between powder particles from each batch, and to determine how those differences affect mechanical properties of AM parts made of those two powder types. Additional discussion of all obtained results is provided.

## 2. Materials and Methods

### 2.1. Powders Utilized in a Test Campaign

The test campaign covered a comparison of two batches of AISI 316L (other specifications UNS S31703 ASTM A240 grade 316L, 1.4404) powder:PAGA powder (Carpenter additive, Widness, UK) produced by Plasma Arc Gas Atomization, specification CT PowderRange 316L F—this powder is one of the most popular on the AM market and widely commercially available in warehouses.UA powder (3D Lab Ltd., Warsaw, Poland). The powder was made of wire, commercially available for TIG/MIG/MAG welding applications—OK Autrod 317LSi (ESAB, Gothenburg, Sweden). The raw material has a low C content, which makes it particularly recommended when there is a risk of intergranular corrosion, and higher Si content, which improves wetting—advantageous in welding as well as here for the atomization process.

UA powder was produced using the ATO Plus system (3D Lab Ltd., Warsaw, Poland), which processes the raw material into powder at a frequency of 35 kHz with the electric arc set up at a current of 110 Amp. This current rating can be considered moderate if compared with a range for typical GTAW welding of stainless steels (100–160 amp for thin items).

The test campaign required 6 kg of each grade of powder, and both powders were in “fresh” conditions as delivered by their suppliers. The particle analysis and chemical composition evaluation of both powders was carried out using the Jeol JSM-6610 scanning electron microscope (SEM) Jeol JSM-6610 (Jeol, Tokyo, Japan) with an energy dispersive spectroscopy (EDS) module—Oxford X-Max.

### 2.2. Powder Property Analysis

Particle size distribution (PSD) was checked on the Keyence VHX-6000 (Keyence International, Mechelen, Belgium) digital microscope and post-processed by binary image processing (with similar quality as the laser diffraction method). Powder flowability depends on grain size distribution. The key properties of the powders for AM applications are tap and apparent densities, and flowability. These properties were measured here with the following methods:Apparent and tap densities—calculated according to the ASTM B212.Flowability—flow rate tested using a calibrated funnel on the Hall flowmeter per the ISO4490:2018 standard.

Both powders were stored in an air-filled container for a few weeks, and they were dried at 150 °C for 60 min before testing.

The chemical composition of the UA powder and its raw material was measured by an inductively coupled plasma-optical emission spectrometry PerkinElmer Optima 4300 DV (ICP-OES method) (PerkinElmer, Inc., Waltham, MA, USA). In the case of PAGA powder, the chemical composition was taken from the quality check datasheet of the purchased powder.

### 2.3. Powder Melting by SLM

Each powder batch was applied to manufacture the 3D test samples. All parts were made on the Laser—Powder Bed Fusion (L-PBF)—based, Selective Laser Melting (SLM) system—the SLM 125HL (SLM Solutions, Lubeck, Germany). To properly select the process parameters dedicated for each powder batch, 33 different laser parameters were taken into account (Table 1 (Part a–c)).

Parameters shown in the first three columns are inputs for calculating the energy density from the Formula (1).
(1)$$\rho_{E}\left[\frac{J}{mm^{3}}\right]=\frac{L_{P}\left[W\right]}{e_{v}\left[\frac{mm}{s}\right]\cdot h_{d}\left[mm\right]\cdot l_{t}\left[mm\right]}$$
where:*L_P_*—laser power [*W*],*e_v_*—exposure velocity [mm/s],*h_d_*—hatching distance [mm],*l_t_*—layer thickness [mm].

During the research, layer thickness and hatching distance were used as the default values across all tested parameter groups, and were equal to 0.03 mm and 0.12 mm, respectively.

### 2.4. Structural Analysis of 3D Printed Samples

The first-level analysis was based on the sample defects by measuring their porosity and pore sizes, which are typical methods for preliminary assessment of melting performance. To select the best parameters group, cubic samples 10 × 10 × 10 mm were manufactured at set-up conditions number 1–33, as in Table 1. To allow proper porosity analysis, samples were mounted in resin, ground with 80, 320, 500, 1200, and 2000 grade abrasive papers, and polished using diamond paste (3 μm grade). Porosity was measured using the KEYENCE VHX-7000 (Keyence international, Mechelen, Belgium) digital microscope. After the process parameter selection, the samples’ microstructures were evaluated. To reveal the microstructure of the samples, and acetic glycerygia solution (6 mL HCl + 4 mL HNO_3_ + 4 mL CH_3_COOH + 0.2 mL glycerol) was applied with an etching time of 20 s. Another structural analysis was based on Vickers hardness distribution measurements using Struers DURA SCAN 70 (Struers, Copenhagen, Denmark) hardness tester.

### 2.5. Tensile and Impact Testing

Samples for tensile and impact testing were manufactured with the use of parameters selected during the structural analysis. Samples were oriented horizontally (the longest dimension—along the samples’ axis was oriented parallel to the substrate plate surface). Axial tensile strength tests were carried out using the Instron 8802 (Instron, Norwood, MA, USA) hydraulic pulsator. An extensometer with a measuring base of 50 mm was used for each tested sample, to allow the best quality data acquisition. During tensile testing, a digital image correlation (DIC) (non-contact, optical method) was used to measure three-dimensional (3D) deformations of the specimen surface by Dantec Dynamics (Dantec, Ulm, Germany) system. The tensile sample was compiled with the ASTM E8/E8M standard; their dimensions are shown in Figure 2.

The impact strength tests of specimens, sized 10 × 10 × 55 mm, were carried out using the Charpy method under the EN ISO 148-1:2010 standard for KCV notch geometry. The Wolpert-Wilson PW 30 (Instron, Norwood, MA, USA) pendulum hammer was applied in the tests. The surface of the specimens was polished before the tests, and all test runs were performed at 20 °C.

## 3. Results and Discussion

### 3.1. Powder Particle Statistical Distribution (PSD)

PSD analysis of PAGA powder was made on the batch in as-delivered condition, i.e., grain size 15–45 μm in the range D10–D90, per production specification. The UA batch output right after production (before classification) was:98.0% of raw material (wire) was converted in powder < 100 μm (i.e., particles useful for AM, melting, coatings), hence, the scrap rate is much lower than in the PAGA process because a conversion rate < 100 μm typically yields in the range of 85–90% [1,3]92.3% of the powder mass can be classified in the range of 20–63 μm, i.e., applicable for SLM, which is also a much better value than for the PAGA process (typically 70–75% for steels)

Eventually, the UA powder was classified as 20–63 μm according to DIN 66165-1 with a sieve shaker brand Multiserw model LPzE-3e at a frequency of 50 Hz. The powder samples for PSD assessment contain 18400 particles for PAGA powder and 8150 for UA powder—both samples had a similar mass, as it follows the fact that PAGA powder was finer than UA. The auxiliary quality metric of the powder batch is also reported: their Span = (D90–D10)/D50 (the lower, the better), and sphericity is defined as the mean value of the proportion of Dim/Dmax for all particles in the batch as both correlated with powder flowability during placement in the melting bed. Figure 3 presents a chart that compares the cumulative distributions of the powder for PAGA and UA methods (at the standard set-up of the ATO system to maximize output). A significant disadvantage of gas atomization is its widespread diameter range [1,3,20]. However, a median particle has a size of 38–45 μm [20], but the upper value at D90 reaches 85–110 μm, with less than 70–75% of the batch (per volume) being a size of <63 μm, which is desirable for AM. The ultrasonic atomization at a frequency of 35 kHz can yield a powder with a median of 44.0 μm; hence 92% of the batch meets PBF requirements at a minimal scrap rate of the rejected powder. Although the UA output (mass rate) is less than that of the PAGA, the ATO system is capable of producing the required powder in a single run with a high conversion rate (>95% of raw material converted in unclassified powder).

In general, the UA powder shows more favorable quality metrics, i.e., lower span, more narrow statistical distribution, and higher sphericity, compared with commercially available PAGA powder. A more limited distribution (i.e., at a low span) and better sphericity of UA powder are apparent in Figure 4. The key statistical parameters are documented in Table 2 (Part A), where the PSD is based on a particle (item) counting, and Table 2 (Part B), where the PSD is evaluated based on the particle set weight. The diameters D10, D50, and D90 are measured as the longest Feret diameters. Comparing the PSD based on optical methods (like binary image processing or laser diffraction), the UA powder seems to be more coarse (median D50 larger by 57%) and more regular (i.e., with Sphericity closer to perfect 1.0), but if the PSD is expressed by volume (mass) fraction, D50 for UA is only 8% higher than that of PAGA. Moreover, the upper values of the PSD (D90) show that the UA batch has fewer large particles than the PAGA batch (also visible in Figure 2 (orange line for UA vs. blue line for PAGA). The additional metric of the powder batches can be a Sauter mean diameter (SMD), which is defined as the diameter of a virtual sphere that has the same volume/surface area ratio for all particles in the batch. This parameter has here practical meaning related to the AM application, because during melting with a laser beam, the heat for melting is transferred mostly through the particle’s outer surfaces, while the enthalpy needed for melting is related to particle mass/volume. So more fine powder has a higher area/volume ratio and can be more easily heated with a laser beam (i.e., can be anticipated to have a higher efficiency of thermal conversion). The SMD value is calculated based on PSD and includes the min-max diameter range for not-perfectly spherical particles (i.e., volume and area proportion is obtained for a sum of all ellipsoids in PSD). The batches here, as used in AM trails, have an SMD value of 35.07 microns for gas-atomized (PAGA) powder, and 46.02 microns for ultrasonic-atomized (UA) powder (after classification < 63 microns). This means that the PAGA batch has 31% less area than the UA batch, for the same batch mass. Nevertheless, one can expect that both batches could require similar laser energy density to melt, as the median mass of a particle is not much different, and the weight share of the large particles in the UA batch is smaller, even if the PAGA batch has a higher area for the energy transfer. Additionally, a balanced particle size distribution should positively influence the maximum packing density, and therefore, the generated component’s density.

Figure 3 also proves that particularly large or small particles are not included in the UA batch. Therefore, a working hypothesis was that PAGA and UA powders, as compared here, would behave similarly during the manufacturing process, which would be validated by the test campaign.

### 3.2. Physical Properties of the Powders

The results of the UA vs. PAGA powders flowability properties are shown in Table 3. The secondary metric is a Hausner ratio between tap and apparent densities, which is preferred to be low. This leads to higher flowability and likely reduced porosity after melting.

The physical parameters for Carpenter powder after tests here were as typically reported for PAGA powders [21]. Based on measurements, one can conclude that UA powder has better flowability (14.83 s/50 g) and higher density. The UA method is supposed to generate fewer gas bubbles in the powder, because the ATO system operates at low pressure (~1 barA), while the brute force makes PAGA powders of the gas stream at high pressure (30–60 barA). Similar observations for UA powder are reported for other materials, e.g., Titanium Grade 5 [6].

### 3.3. Chemical Composition

The results of the UA powder’s chemical composition compared to PAGA powder are listed in Table 4. The variations in the composition were small, and the powder meets the criteria of the AISI 316L specification. More specifically, there was a minor reduction of the manganese (Mn) content from 1.6 to 1.0 %. This change is caused by the minor evaporation of Mn, which is faster than for other elements at atomization temperature, and is resulted in an ATO system from trade-off. The silicon (Si) content in UA powder was slightly higher than the standard AISI 316L specification, but this is the result of the raw material selection, which is a welding wire type 315LSi (here max Si content up to 0.90% per supplier specification). Anyway, the difference in Si content between PAGA and UA batches is negligible here. Another monitored parameter for many powder grades is oxygen (O) content. This value is usually not stated for stainless steel, but a typical 316L powder from gas atomization has 0.05–0.10% of oxygen [21].

### 3.4. Particle Morphology

The particles presented in Figure 4 show a more spherical shape of UA powder from the ATO system (right picture), which is required by AM market, and is indicative of good flowability and a high packing density of the powder (as proven by tests). Furthermore, the SEM picture of UA does not show agglomerates, deformed, and satellite particles, which are common for gas atomization (left picture). The median UA particles are more significant than the PAGA batch due to classification in the ranges of 20–63 μm and 15–45 μm, respectively.

Based on the obtained images, there are visible differences between the two powders. PAGA powder is characterized by a uniform structure with moderately spherical shapes and some satellites on the grains. In the case of UA powder, the particles are more spherical, with some spots visible on their external surface. Some particles have been subjected to deeper analysis to determine the chemical composition in the area of the mentioned spots. The results of the chemical composition analysis of one of the spots are shown in Figure 5. One should note that the surface precipitations have a very small size (<5 μm) and thickness (<2 μm), hence the negligible effect on the melted product properties, while the bulk chemical composition of the batch is fulfilling the AISI 316L specification.

Small brighter spots on the particle’s surfaces exhibit increased Cr and Mn content based on the EDS measurements. One hypothesis of the segregation of such elements is related to the lower weight of the mentioned elements compared to Fe and Ni which also have a significant share in the material structure. The EDS analyses were made on UA powder samples to determine if such a phenomenon occurs after the AM process. The results exposed no structural unevenness. Additionally, the powder particles taken from both powder batches were mounted and etched—see Figure 6. It was no registered increased internal porosities in both powder samples. The microstructures of both powder types are characterized by regular grains without any significant imperfections.

### 3.5. Structure of AM Parts

Thirty-three cubic samples 10 × 10 × 10 mm for each powder batch were additively manufactured with modified parameters, as shown in Table 1. Porosity and defect measurements were taken on the cut surfaces with all layers visible-yz plane (where xy is a substrate plate surface and the z-axis characterized height during the AM process). The obtained results are shown in Figure 7 for porosity vs. the energy density $\rho_{E}$ and Figure 8 for the maximum defect vs. the energy density $\rho_{E}$.

Regarding the obtained results, it could be stated that, by using low exposure velocity (700 mm/s), the porosity in samples made of PAGA powder is higher than in the same UA parts manufactured using the same process parameters. After reaching 800 mm/s, the differences in porosity and defect sizes are no different than for typical values of AM process variability. In the case of higher values of exposure velocity (900 mm/s), the porosity increased because of the generation of void related to the “lack of fusion” phenomenon. The primary condition as defined parameters in tensile tests was a minimal porosity value. Based on that rule, the chosen parameters for each powder batch are shown in Table 5. The No. 19 group and No. 22 group were used for the manufacturing process of samples dedicated to tensile testing and impact strength analysis.

The selected printing parameter means that the UA material was melted at almost 15% higher energy density than PAGA (laser power increased by 30 W). This is not surprising if one considers that the PAGA powder has a 31% higher area/volume ratio than the UA batch, as PAGA powder is finer. Because during melting with a laser beam, the heat is transferred through the particle outer surfaces, it is likely to reach higher thermal conversion efficiency for PAGA powder, and is observed to have a lower energy density parameter. However, higher energy doses during melting affect the microstructure in the samples, shown in Figure 9, and strength properties, as discussed later. Some visible differences in the AM parts microstructure are not associated directly with material batch PAGA vs. UA or their composition/microstructure, but rather how intensely the samples were melted with aid of a laser beam [15,22,23,24,25]. Most likely the differences in energy density and microstructure shown in Figure 9 could be completely eliminated if UA was either made finer (i.e., at a higher ultrasonic frequency) or classified as finer.

The additional structure-based test allowed for the determination of material hardness with the HV 0.5 method. The results are shown in Table 6. Samples obtained using UA powder and 8% lower hardness values characterize higher energy density, which is typical behavior during exposure of the powder to the higher energy density in stainless steels. Our previous research [26,27] deliberated such behavior during a deep analysis of the layered structure in AM samples made from the AISI 316L steel. It was proven that very local heat treatment from a laser beam creates a specific cooling effect between consecutive layers (from heat dissipation to deeper layers), which influences the phase changes and the hardness [28], as measured by a sclerometer. The measured thickness of layers in the PAGA samples is smaller than in the same parts produced using UA powder in samples obtained in the current research.

In the case of the AM 316L steel, there is also a phenomenon related to the influence of used higher energy density. Delivering a significant amount of energy into the material’s volume causes an increase in the precipitation effect, which affects the drop in the hardness value [13,22,23].

### 3.6. Tensile Tests with DIC Analysis

The dog-bone-shaped specimens were produced in the L-PBF process according to the ASTM E46696 standard to obtain good quality results, and therefore all specimens were ruptured in the middle of the testing zone. The obtained values of the tensile test are shown in Figure 10, and the most important parameters were set together in Table 7.

All samples significantly exceeded the standard AISI 316L specifications, which are:ultimate tensile strength UTS > 510 MPa,0.2% offset yield strength R0.2 > 350 MPa,elongation at break > 30%,Charpy impact resistance KCV > 80 J/cm^2^.

One observation from the tensile test was the lack of a visible yield limit point in all samples. Samples made of UA powder represented slightly reduced (by 1.5%) ultimate tensile strength (UTS) and 0.2% offset yield strength (R_0.2_), along with significantly larger (by 9.4% point) elongation at break (column “Elong.” in Table 7).

The phenomenon of increased elongation at break makes the UA-powder samples more plastic, but at the same time, they are at a slightly reduced strength compared to the as-built PAGA-powder samples. The Young modulus in UA-based samples is visibly higher (about 10%) compared to the PAGA-based samples. Such a phenomenon was analyzed by Niendorf et al. [29] and Rottger et al. [30], where it was described an influence of used energy density on Young Modulus; it increases with the increase of used energy density. The above-mentioned issues could be better understood while considering the DIC analysis, shown in Figure 11. Such analyses were made to detect any unevenness during the tensile testing process. Additionally, the material behavior at some characteristic stages could be revealed as UTS or yield point (R_0.2_ in Figure 11). In the case of the UA-powder-made samples, a significantly bigger area of the high strain level is visible, compared to the PAGA-powder-made samples. Such results could be useful for identifying some fatigue properties of the material as shear of elastic or plastic areas during the loading process.

DIC analysis revealed different behaviors for each material during strain. Samples made of PAGA powder deformed less equally in a smaller area than samples made of UA until reaching the UTS, where there were visibly increased strain areas in PAGA samples. In both cases, necking is visible, proving the high plasticity of test parts made of PAGA and UA powders.

The difference in tensile properties for samples made of PAGA vs. UA powders can be explained by the fact that, in the case of the UA powder, the selected process parameter (#22 as in Table 5) represented a 15% higher energy density than for PAGA powder (#19 as in Table 5). Previous research by the authors [11,16,27] evaluated this phenomenon for samples made of the same powder (only from PAGA at that time) and adjusted laser parameters. The current and previous [26] test results can be plotted together on one chart (Figure 12) to better observe the effect of the laser energy density on properties like UTS, R0.2, and the elongation at break. The values for UA powder marked on the chart with red marks fall between the other values for larger and smaller energy density collected from other tests for PAGA powder (from the same supplier and test methodology). The general trend observed in all tests is that UTS, R_0.2,_ and elongation values drop after the laser energy increase above 60 J/mm^3^. So it is evident that the differences in strength properties between UA-#22 samples and PAGA-#19 samples can be mostly attributed to melting conditions, not to properties of the bulk powder.

To better describe two tested materials’ behavior, fracture images of selected tensile samples were made (Figure 13).

Both of the tested samples indicate a mixed character of cracking—mostly plastic, but with local features of brittle-like cracking. Samples made of UA powder indicate more areas of brittle cracking, when at the same time, in the samples made of PAGA powder, there are more voids visible. Such porosity (Figure 13A) could be related to lower energy density used for manufacturing, which caused local lack of fusion between some powder particles.

### 3.7. Impact Resistance

The last part of the test campaign was a Charpy impact test. As performed in tensile testing, five samples were manufactured using each material. The obtained results are shown on the chart in Figure 14.

In average values, samples made of UA reached impact energy equal to 121.0 J, whereas at the same time, PAGA samples reached an average of 148.4 J, differing by about 20%. Such a phenomenon could be related to the decreased hardness of the material in samples made of UA powder, which makes the material more plastic.

Also, the effect shown in Figure 12 and described above can be a plausible explanation for impact resistance reduction on UA powder. Nevertheless, all samples easily satisfy the AISI 316L Si specification, which requests that the Charpy impact resistance KCV reach at least 80 J/cm^2^ at room temperature.

## 4. Conclusions

The quality and performance of any 3D printed parts strongly depend on the quality of the powder used in the AM process. However, a preliminary comparison of PAGA vs. UA powder batches allows for the conclusion that UA powder has similar quality as PAGA powder, and can likely be considered to be equivalent. Additionally, it is possible to reach similar AM part properties by properly selecting the production process parameters. Based on the obtained research results, the following conclusions could be drawn:In the case of UA powder, 92.3% of the whole material mass can be classified in the range of 20–63 μm, which is a much bigger value than for the PAGA process (typically 70–75% for steels).The UA material needs almost 15% higher energy density than PAGA (laser power increased by 30 W).Reduced ultimate tensile strength (from 621 MPa to 612 MPa) with higher elongation at break (from 36% to 41%) characterizes samples melted of UA powder utilizing the selected parameters compared with PAGA, which gives obtained values closer to the conventionally made material.The Young’s modulus in UA-based samples is visibly higher (about 10%) compared to the PAGA-based samples, which increases with the increase of used energy density.A 20% reduction of the impact strength in the case of UA-made samples was registered, which could be related to the increased plasticity of the material.

## Figures and Tables

**Figure 1 materials-15-04938-f001:**
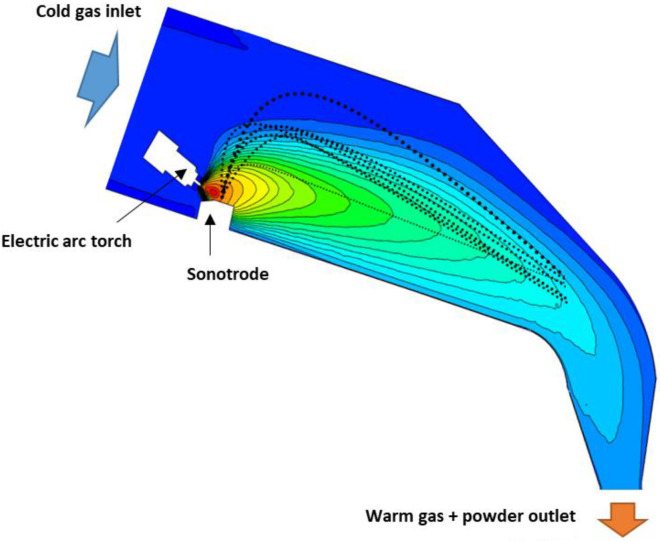
Atomization chamber with the temperature profile of the gas and droplet trajectories during the atomization.

**Figure 2 materials-15-04938-f002:**
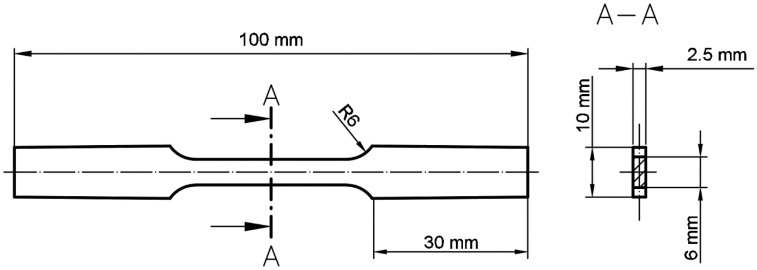
Dimensions of the ASTM E8\E8M standard tensile test dog-bone sample.

**Figure 3 materials-15-04938-f003:**
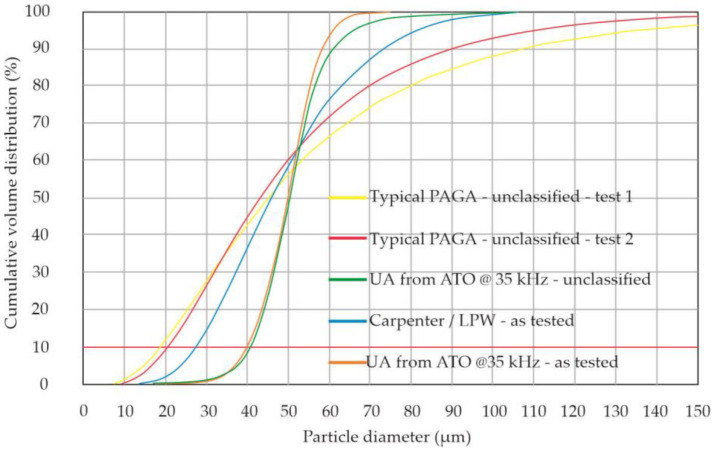
Cumulative statistical distribution (volume %) for the powder batches as in the test campaign, test 1 and test 2 courses were taken from the references [1,3] respectively.

**Figure 4 materials-15-04938-f004:**
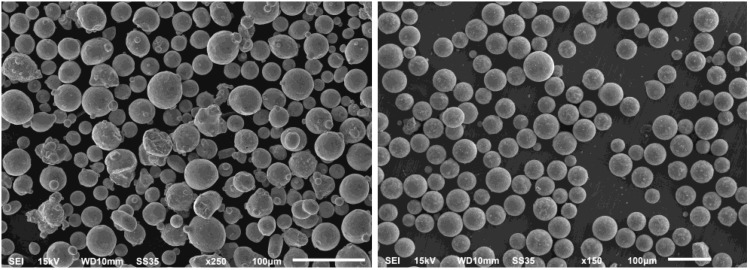
SEM scans of PAGA (**left**) and UA (**right**) particles as tested—the same scale.

**Figure 5 materials-15-04938-f005:**
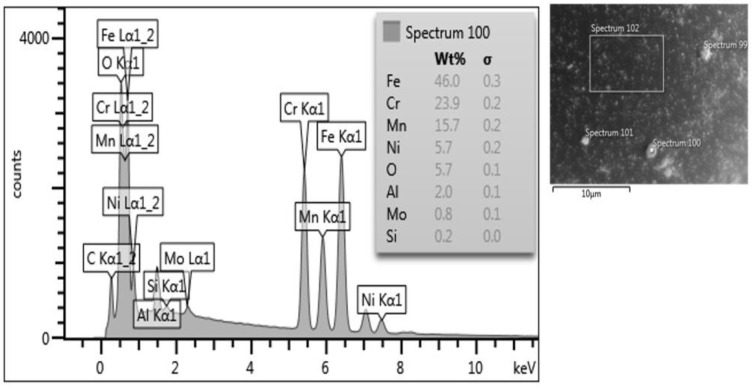
The EDS tests results of one of the spots in the 316L grain surface.

**Figure 6 materials-15-04938-f006:**
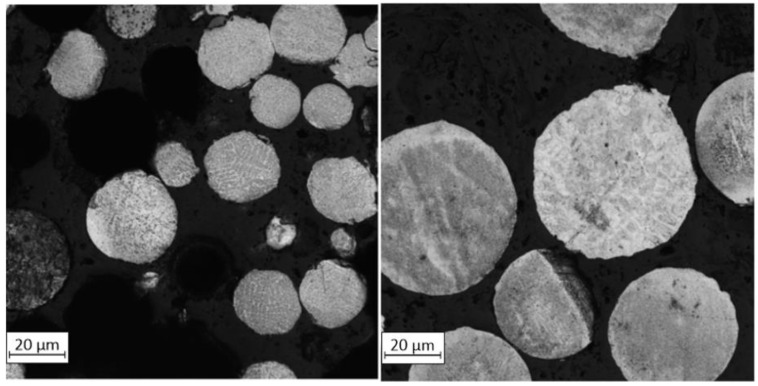
Microstructure of both powders particles: **left**—PAGA; **right**—UA.

**Figure 7 materials-15-04938-f007:**
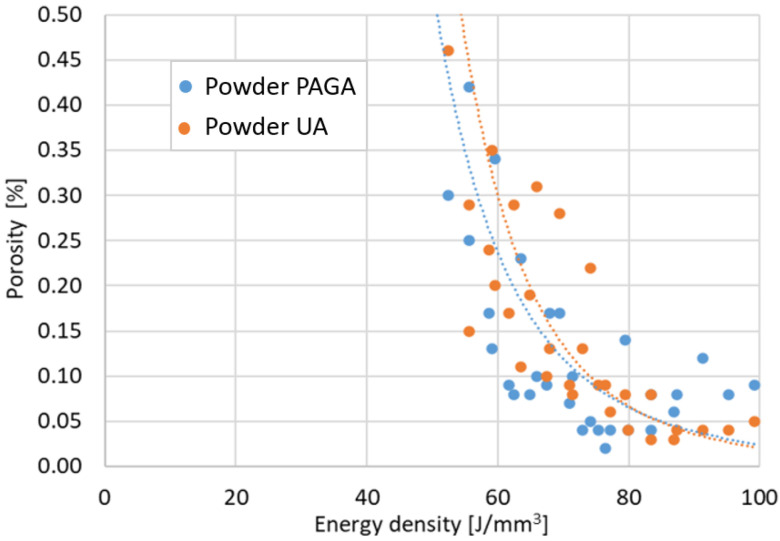
Sample porosity in the function of the laser energy density $\rho_{E}$.

**Figure 8 materials-15-04938-f008:**
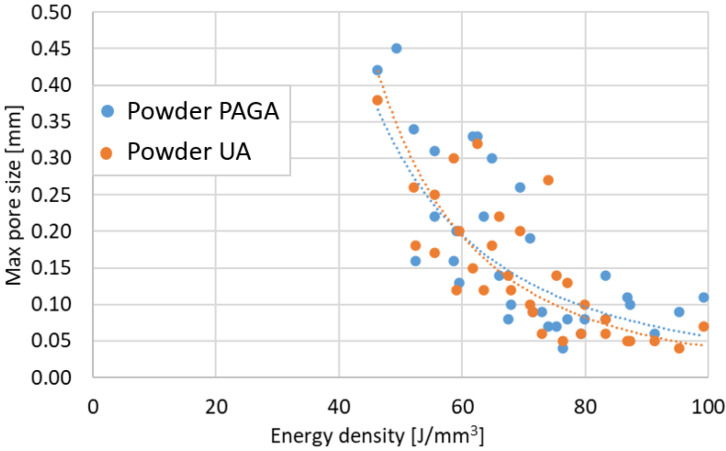
Maximum defect sizes in the function of the laser energy density $\rho_{E}$.

**Figure 9 materials-15-04938-f009:**
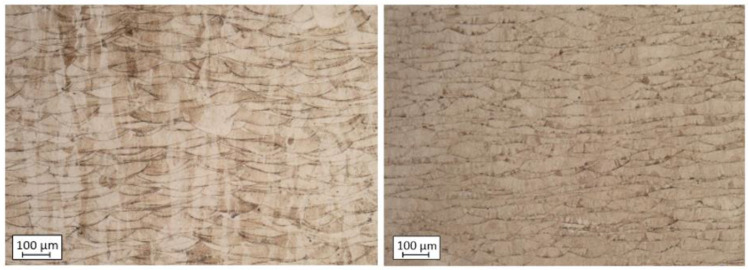
Microstructure of samples manufactured using both powders: **left**—PAGA; **right**—UA.

**Figure 10 materials-15-04938-f010:**
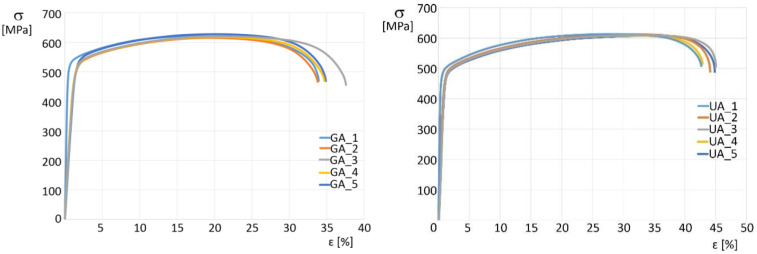
Stress—strain chart of tensile samples made of powders: **left**—PAGA; **righ**—UA.

**Figure 11 materials-15-04938-f011:**
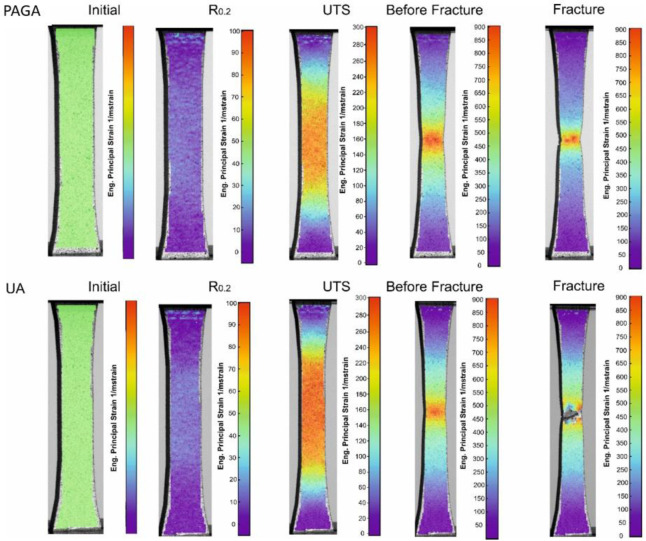
DIC results of samples manufactured using both powders during tensile testing.

**Figure 12 materials-15-04938-f012:**
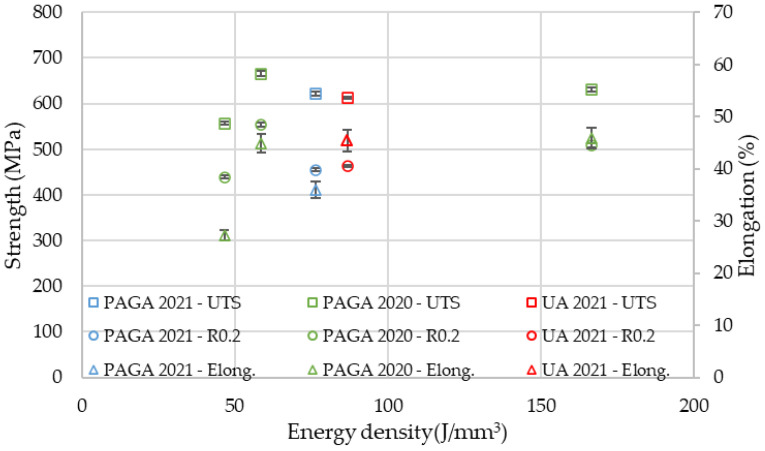
PAGA and UA powder sample strength and elongation vs. laser energy density (data PAGA-2020 from previous own paper [26]).

**Figure 13 materials-15-04938-f013:**
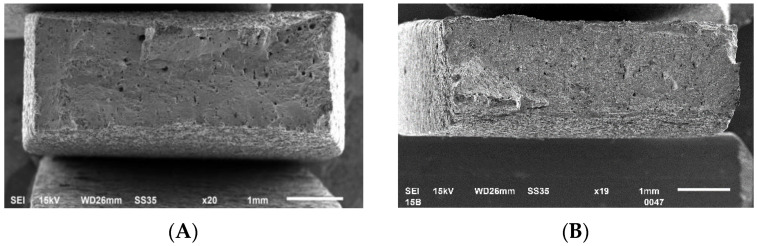
Fracture images of samples’ subjected to tensile testing: (**A**)—PAGA; (**B**)—UA.

**Figure 14 materials-15-04938-f014:**
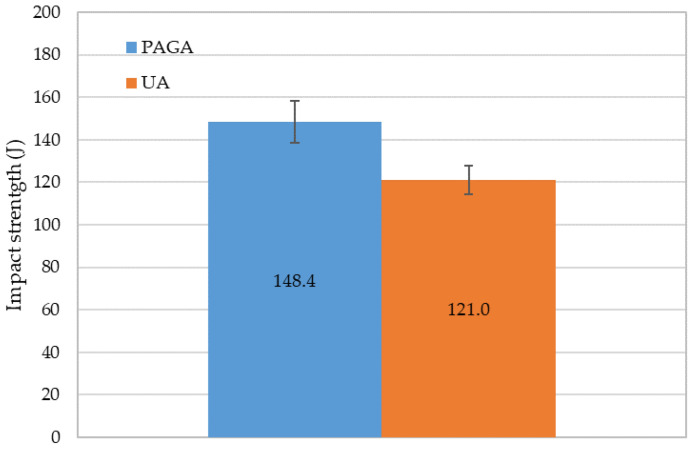
Impact strength of samples manufactured using both powders.

**Table 1 materials-15-04938-t001:** a. Sets of analyzed process parameters for exposure velocity equal to 700 mm/s. b. Sets of analyzed process parameters for exposure velocity equal to 800 mm/s. c. Sets of analyzed process parameters for exposure velocity equal to 900 mm/s. c. Sets of analyzed process parameters for exposure velocity equal to 900 mm/s.

a.		
No.	*L_P_* [*W*]	$\rho_{E}$ [J/mm^3^]
1	150	59.52
2	160	63.49
3	170	67.46
4	180	71.43
5	190	75.40
6	200	79.37
7	210	83.33
8	220	87.30
9	230	91.27
10	240	95.24
11	250	99.21
**b.**		
**No**.	***L_P_* [*W*]**	$\rho_{E}$ **[J/mm^3^]**
12	150	52.08
13	160	55.56
14	170	59.03
15	180	62.50
16	190	65.97
17	200	69.44
18	210	72.92
19	220	76.39
20	230	79.86
21	240	83.33
22	250	86.81
**c.**		
**No.**	***L_P_* [*W*]**	$\rho_{E}$ **[J/mm^3^]**
23	150	46.30
24	160	49.38
25	170	52.47
26	180	55.56
27	190	58.64
28	200	61.73
29	210	64.81
30	220	67.90
31	230	70.99
32	240	74.07
33	250	77.16

**Table 2 materials-15-04938-t002:** A. PSD of the powder batches as in the binary image processing (particle counts). B. PSD of the powder batches based on the weight fractions.

A.		
Production Method	PAGA	UA
Brand	Carpenter/LPW	3D Lab/ATO
D10 [μm]	16.4	34.1
D50 [μm]	29.4	46.1
D90 [μm]	50.4	55.2
Span = (D90 − D10)/D50	1.078	0.476
Average Sphericity	0.75	0.89
**B.**		
**Production Method**	**PAGA**	**UA**
Brand	Carpenter/LPW	3D Lab—ATO
D10 [μm]	27.3	40.9
D50 [μm]	45.8	49.7
D90 [μm]	73.1	58.2
SMD [μm]	35.1	46.0

**Table 3 materials-15-04938-t003:** Physical properties of the powder batches.

Production Method	PAGA	UA
Brand	Carpenter/LPW	3D Lab/ATO
Sieve classification [µm]	15–45	15–60
Tap density [g/cm^3^]	4.67	4.72
Apparent density [g/cm^3^]	4.29	4.40
Hausner ratio	1.089	1.072
Flow rate [s/50 g]	18.69	14.83

**Table 4 materials-15-04938-t004:** Chemical composition of PAGA and UA powders.

Element (wt.%)	Specification AISI 316L	Carpenter—LPW CT Powder Range 316L F	3D Lab—ATO System Wire→UA Powder	
Fe	Balanced	Balanced	Balanced	Balanced
C	<0.030	0.027	0.015	0.010
Cr	16.0–18.0	17.8	17.4	17.3
Cu	-	0.02	0.14	0.11
Mn	<2.0	0.98	1.6	1.0
Mo	2.0–3.0	2.31	2.4	2.4
N	<0.10	0.09	0.095	0.040
Ni	10.0–14.0	12.8	10.5	10.5
O	-	0.02	-	0.022
P	<0.045	0.011	0.025	0.025
S	<0.03	0.004	0.007	0.006
Si	<0.75	0.72	0.77	0.76

**Table 5 materials-15-04938-t005:** Parameters selected for tensile samples.

No.	Powder Batch	*L_P_* [*W*]	*e_v_* [mm/s]	*h_d_* [mm]	*ρ_E_* [J/mm^3^]	Porosity [%]	Pore Size [mm]
19	PAGA	220	800	0.12	76.39	0.02	0.04
22	UA	250	800	0.12	86.81	0.03	0.05

**Table 6 materials-15-04938-t006:** Hardness measurements of samples obtained using PAGA and UA powders (#19 and #22 are process parameters from Table 5).

Powder Sample	Value [HV]	Standard Deviation [HV]
PAGA #19	233.67	3.55
UA #22	216.44	3.65

**Table 7 materials-15-04938-t007:** The results registered during tensile testing of samples obtained using PAGA and UA powders.

	PAGA Powder				UA Powder			
Sample no.	UTS [MPa]	R_0.2_ [MPa]	E [GPa]	Elong. [%]	UTS [MPa]	R_0.2_ [MPa]	E [GPa]	Elong. [%]
1	621.29	452.65	149.66	34.16	615.91	432.27	167.75	42.96
2	616.65	454.55	150.22	34.03	614.83	437.45	166.22	47.39
3	621.52	452.12	150.44	37.85	612.35	437.42	158.44	46.55
4	618.50	452.27	149.46	34.59	610.56	437.62	162.23	43.09
5	628.65	460.89	149.69	34.31	609.49	437.86	160.47	45.75
Average	621.14	454.78	150.68	35.98	612.68	436.45	163.14	45.41
Standard deviation	4.56	3.78	0.43	1.61	2.58	2.45	3.91	2.03

## Data Availability

Not applicable.

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
