# Peer review of "A Comparative Study on Laser Powder Bed Fusion of Differently Atomized 316L Stainless Steel"

_materials, 2022, doi:10.3390/ma15144938_

Round 1

Reviewer 1 Report

Paragraph 2.4: It seems that the specimens used for the tensile and impact testing are manufactued similarly to the cubic sample in §2.3. For the sake of clarity it should be mentioned that it was so.

Line 243: To be checked: numerical value written in a confusing manner to express a range

Line 270: Do you mean Fig. 4 ?

Line 278: Please check and correct figure number ! (Presumably figure 3)

Line 291: Missing word: more or less? Please check and correct the sentence !

Line 298: (a) Figure 3 or figure 4? (b) You may want to temper your affirmation since the picture of UA particles in Fig. 4 shows small particles.

Line 342: Do you mean 20 to 63 microns and 15 to 45 microns ? Please check or explain !

Line 381: (a): Check the English of the sentence ! (b): What about the  900 mm/s velocity?

Lines 405-414: You may want to refer to Table 6 in the text at the proper place !

Line 446: Please check: from Table 7 it is the opposite and contradicts the affirmation in lines 448-449 !

Line 473: tests instead of testes

Line 549: You may want to check with the editor if a mention about the fact that 2 authors belong to the 3D-Lab manufacturing the UA powders. Should this fact be mentioned again in this section?

Author Response

Dear Reviewer,

On behalf of all authors, I would like to thank you for taking your time to read our manuscript and put your comments which allowed us to improve the quality of our work. Below you can find our answers related to each of your comments.

  1. Paragraph 2.4: It seems that the specimens used for the tensile and impact testing are manufactued similarly to the cubic sample in §2.3. For the sake of clarity it should be mentioned that it was so.

Ad.1. We put proper description in the mentioned paragraph:

“Samples for tensile and impact testing were manufactured with the use of parameters selected during the structural analysis. Samples were oriented horizontally (the longest dimension – along the samples’ axis was oriented parallel to the substrate plate surface).” It has been yellow-highlighted.

Also, we mentioned process parameters for samples dedicated to tensile and impact testing that was produced, it is placed below fig 8 and yellow-highlighted. We put the following sentence (it was yellow- highlighted). “No. 19 group and No. 22 group were used for the manufacturing process of samples dedicated to tensile testing and impact strength analysis.”

  1. Line 243: To be checked: numerical value written in a confusing manner to express a range

Ad.2. Thank you for that comment, this issue has been corrected

  1. Line 270: Do you mean Fig. 4 ?

Ad.3. Yes, we meant fig. 4 – it is correct now.

  1. Line 278: Please check and correct figure number ! (Presumably figure 3)

Ad.4. Thank you for that comment, this issue has been corrected

  1. Line 291: Missing word: more or less? Please check and correct the sentence !

Ad.5. The sentence has been corrected.

  1. Line 298: (a) Figure 3 or figure 4? (b) You may want to temper your affirmation since the picture of UA particles in Fig. 4 shows small particles.

Ad.6. Figure 3 is correct. In this sentence, we mentioned particles that are above the limit (20 – 63 microns). Particles in figure 4 were selected in a stochastic way, only to allow its shape description.

  1. Line 342: Do you mean 20 to 63 microns and 15 to 45 microns ? Please check or explain !

Ad. 7. Range 20 – 63 microns is a range for PAGA (from LPW company) 15 – 45 microns is a range for UA (from 3D Lab company). Those two ranges were mentioned to justify the differences in median value.

  1. Line 381: (a): Check the English of the sentence! (b): What about the 900 mm/s velocity?

Ad.8. (a) The mentioned sentence was corrected. (b) we put an additional comment in the text: “In the case of higher values of exposure velocity (900 mm/s), the porosity increased because of the generation of void related to the “lack of fusion” phenomenon”

  1. Lines 405-414: You may want to refer to Table 6 in the text at the proper place !

Ad.9. Thank you for that comment – we put a proper reference in the text. It was yellow-highlighted.

  1. Line 446: Please check: from Table 7 it is the opposite and contradicts the affirmation in lines 448-449 !

Ad.10. It was a mistake. It should be a word “higher” there. Now it is correct. Thank you for that comment.

  1. Line 473: tests instead of testes

Ad.11. Thank you for that comment, this issue has been corrected

  1. Line 549: You may want to check with the editor if a mention about the fact that 2 authors belong to the 3D-Lab manufacturing the UA powders. Should this fact be mentioned again in this section?

Ad.12. We have written this manuscript based on the proper template. The affiliation of two authors from the 3DLab company is mentioned in the beginning. This manuscript is a result of our cooperation. Of course, we could add some more data if an editor would ask for it – for us, it is not a problem.

Reviewer 2 Report

This paper reported comparisons of powder properties, microstructure and mechanical strength of SLM process between powder productions by Ultrasonic Atomization (UA) and Plasma Arc Gas Atomization (PAGA). Experimental procedure and obtained results are explained in detail in this article.

It is considered that this paper has low scientific soundness because considerations for the different results between UA and PAGA processes seem to be insufficient. However, some valuable results were presented in point of engineering view. These experimental results will be useful for AM researchers and engineers.

I recommend that this paper be accepted in the present style. However, there are a few mandatory revised points as follows:

1. Temperature scale must be put in Fig.1

2. ρE in Eq.(1) is rewritten as QE

Author Response

Dear Reviewer, 

We would like to thank you for your polite review. In the case of your first comment, we were not able to put that scale, because it is a general scheme applicable for all metals which have different melt temperatures. 

In the case of the rho symbol, all issues have been corrected. 

Reviewer 3 Report

I must appreciate the efforts of authors to conduct this research on a comparison between Gas Atomization (GA) and Ultrasonic Atomization (UA) and it is timely done. However, there are some observation that may be addressed:

1. This is a lengthy manuscript with lot of results which compares both GA and UA. The authors must report those results which are really comparable with reasoning instead of presenting all of them. This may lead to reduce the length of this paper.

2. Novelty of this manuscript is required to be clarified.

3. Conclusions do not justify which of both GA and UA is better.

4. There are few instances where self citations are present 

5. The levels of process parameter in Table 1 must be written in a separate table.

6. Layer thickness was kept constant to be 0.03 mm. Well, hatching distance seems also constant with value 0.12 mm which can be removed from table with a similar sentence replacement as done for layer thickness.

Author Response

Dear Reviewer,

On behalf of all authors, I would like to thank you for taking your time to read our manuscript and put your comments which allowed us to improve the quality of our work. All corrections made based on your review are blue-highlighted. Below you can find our answers related to each of your comments:

  1. This is a lengthy manuscript with lot of results which compares both GA and UA. The authors must report those results which are really comparable with reasoning instead of presenting all of them. This may lead to reduce the length of this paper.

    Ad.1. Results that we were shown in our manuscript were not randomly chosen. All our research was designed to provide all necessary data to allow objective comparison of PAGA and UA technologies. There are not any available standards that would characterize some specific testing methods of powders dedicated to additive manufacturing or samples obtained with the use of such powders. We could not point out more or less necessary results to remove them.  That is why we cannot remove it. We hope you will understand our point. 
  2. Novelty of this manuscript is required to be clarified.

    Ad.2. Near the end of the introduction we put the aim of our manuscript - it has been blue-highlighted.
  3. Conclusions do not justify which of both GA and UA is better.

    Ad.3. The aim of our research was not to point to better powder but only to compare them. It is very hard to point out which powder is better.

    An example: PAGA technology is better than UA for high-volume production, while UA is better than PAGA for low-volume production. Big companies have a much bigger requisition for significant volumes of powder (mostly one type) than universities or small companies. 

  4. There are few instances where self citations are present 

    Ad. 4. Self-citations are allowed in the Materials journal. Additionally, we used some results from the past in our comparison so we have to cite such results.
  5. The levels of process parameter in Table 1 must be written in a separate table.

    Ad.5.  The table has been divided into three parts (for each exposure velocity level)

  6. Layer thickness was kept constant to be 0.03 mm. Well, hatching distance seems also constant with value 0.12 mm which can be removed from table with a similar sentence replacement as done for layer thickness.

    Ad.6. We totally agree with your comment. Proper corrections were made.